# HEalth And Dementia outcomes following Traumatic Brain Injury (HEAD-TBI): protocol for a retrospective cohort study

Emma Rosalyn Russell [1], Donald M Lyall,[2] William Stewart [1,3]

¹School of Psychology and Neuroscience, University of Glasgow, Glasgow, UK
²School of Health and Wellbeing, University of Glasgow, Glasgow, UK
³Neuropathology Research Laboratory, NHS Greater Glasgow and Clyde, Glasgow, UK

**Correspondence to**
Dr Emma Rosalyn Russell;
emma.russell@glasgow.ac.uk

## ABSTRACT

**Background** It is estimated that by 2050 the global incidence of dementia will have exceeded 152 million. At present, there are no effective therapies for dementia, with a focus in research now turning to strategies for disease prevention. Traumatic brain injury (TBI) is recognised as a major risk factor for dementia; estimated to be responsible for at least 3% of cases in the community. However, adverse health outcomes after TBI are not restricted to dementia. A wide range of conditions are documented among TBI survivors, many of which also increase dementia risk. 'HEalth And Dementia outcomes following Traumatic Brain Injury' is a study aiming to explore the hypothesis that increased dementia risk following TBI reflects both the direct effect of the injury on the brain and the indirect effects of wider, adverse health outcomes associated with TBI which, in turn, increase dementia risk.

**Methods and analysis** Comprehensive electronic medical and death certification records will be analysed for individuals with a documented history of TBI, compared with those of a matched general population control cohort with no documented TBI exposure. Cox proportional hazard regression models will be run to compare outcomes. Furthermore, existing diagnostic imaging and radiological reports for the cohort will be analysed to identify evidence of specific white matter abnormalities in TBI exposed individuals and their controls, and establish their potential diagnostic utility.

**Ethics and dissemination** Approvals for the study have been obtained from the University of Glasgow College of Medical, Veterinary, and Life Sciences Research Ethics Committee (project number 200220038) and from National Health Service Scotland's Public Benefits and Privacy Panel (application 2122-0224). As results emerge, these will be presented at appropriate multidisciplinary research conferences and made available through open access platforms where possible.

## STRENGTHS AND LIMITATIONS OF THIS STUDY

⇒ Using a comprehensive dataset of individuals with a history of traumatic brain injury compared with matched population controls— matched on sex, year of birth and socioeconomic status—minimises risks of bias.

⇒ It is known that there are limitations in the granularity of the data—brain injury data will be coded in accordance with the International Classification of Diseases, we will not have access to self-reported information or general practitioner/hospital notes.

⇒ Errors in health record coding are widely recognised, but are assumed to be consistent between our cohort of interest (those with a history of traumatic brain injury) and their matched population controls.

⇒ The National Health Service was not founded until 1948, and we therefore may have less complete medical records for individuals born in the early 1900s.

## BACKGROUND

Over a quarter of a million people in the UK are living with a dementia diagnosis, with an annual cost to the economy of around £34 billion.[1] Despite decades of research, we have no effective therapies for dementia. In part, this is a consequence of the disease having been active for many years by the time a diagnosis is made, by which time the neurodegenerative process is both established and irreversible. Attention, therefore, has begun to focus on developing strategies to prevent or delay the onset of disease. Reflecting this, dementia prevention is regarded as a major public health priority by the WHO.[2] Further, the 2021 Alzheimer's Innovation Readiness Index, coauthored by Alzheimer's Disease International and the Global Coalition on Aging, calls for national campaigns to address dementia risk factors and for investment in research to better understand their contribution to disease.[3]

The 2020 Lancet Commission on Dementia, Intervention and Care identifies 12 potentially modifiable risk factors for dementia.[4] Among these, traumatic brain injury (TBI) is estimated to contribute to in excess of 3% of dementias in the community. Importantly, whereas for many, if not all other dementia risk factors, the temporal relationship between risk of exposure and onset of

neurodegenerative process is unknown, for TBI the time of injury is known (because it is a discrete, single 'event'). As such, studies in patients with TBI provide unique opportunity to track the development of disease over time, with potential to identify novel strategies for early diagnosis and intervention. However, while the association between TBI and neurodegenerative disease risk is recognised,[5–8] this is not the only adverse health outcome associated with TBI.[9–11] In addition to increased risk of neurodegenerative disease, individuals with history of TBI are also recognised as having earlier mortality than expected, with increased risk of many common diseases such as cardiovascular disease, cancer and respiratory diseases.[12] As such, increased risk of neurodegenerative disease in those surviving TBI might reflect the direct effect of the injury to the brain and/or a consequence of wider adverse health outcomes contributing to increased dementia risk.

HEalth And Dementia outcomes following Traumatic Brain Injury (HEAD-TBI) is specifically designed to assess the complex interaction between TBI and lifelong health outcomes contributing to dementia risk following injury. In so doing, HEAD-TBI will provide novel insights into the relationship between TBI and neurodegenerative disease risk with potential to inform strategies to identify those at greatest risk of adverse outcome and possibilities for intervention to mitigate risk. In addition, by providing unique opportunity to study trajectory of disease from initiation to outcome, HEAD-TBI will have potential wider application to research in non-TBI associated dementias. The study will run for a duration of 3 years, during October 2022 to October 2025.

## Aims

The specific aims of HEAD-TBI are to:

▶ Explore the risk of dementia and wider neurodegenerative diseases in individuals with a history of TBI compared with matched general population controls.

▶ Compare causes of death and wider health outcomes, including mental health, in individuals with a history of TBI compared with matched general population controls.

▶ Assess the relationship between structural brain imaging differences and risk of dementia following TBI.

## METHODS AND ANALYSIS

Using established protocols that have been proven highly successful in addressing similar research aims in other populations of interest,[13 14] we now propose conducting a retrospective cohort study looking at a wide range of physical and mental health outcomes, including neurodegenerative disease, in individuals exposed to TBI, compared with an age, sex and social deprivation matched general population comparison group.

**Table 1** ICD10 codes for traumatic brain injury

| | |
|---|---|
| S06.0–S06.9 | Concussion<br>Traumatic cerebral oedema<br>Diffuse brain injury<br>Focal brain injury<br>Epidural haemorrhage<br>Traumatic subdural haemorrhage<br>Traumatic subarachnoid haemorrhage<br>Intracranial injury with prolonged coma<br>Other intracranial injuries<br>Intracranial injury, unspecified |
| S07.1, S07.8, S07.9 | Crushing injury of skull<br>Crushing injury of other parts of the head<br>Crushing injury of head, unspecified |
| S09.7–S09.9 | Multiple injuries of head<br>Other specified injuries of head<br>Unspecified injury of head |

## Health record interrogation

Individuals with a documented history of TBI will be our cohort of interest and will be identified from records held by the electronic data research and innovation service (eDRIS), who are part of Public Health Scotland, and hold a wide range of health information, including prescription information and hospital attendance, as well as death certification data for residents in Scotland. Individuals who have prior TBI recorded (coded as International Classification of Disease 9/10 (ICD 9/10)) in their medical records (namely hospital attendance, Scottish Morbidity Record (SMR) 01) will be selected. Individuals will be considered to have a documented TBI if their hospital records contain one of the ICD9/ICD10 codes shown in tables 1 and 2). A wide range of TBI severities are listed in the chosen codes, ranging from those with a history of mild TBI such as concussion, to those with more moderate-to-severe brain injuries. This will enable for broad analyses covering all TBI outcomes, to more stratified analyses investigating lifelong health outcomes following certain types or severities of injury. Community Health Index (CHI) numbers will be used to link individuals to their own electronic health record data. The CHI is a 10-character code, which uniquely identifies each individual in the National Health Service (NHS) Scotland registry.

Matched population controls for the study cohort will be selected on a 3:1 basis, that is, for every one individual with exposure to TBI, three matched controls will be selected. This aims to minimise any bias that may be present, and to increase study power, which can aid the detection of a statistically significant difference in outcome variables.

**Table 2** ICD9 codes for traumatic brain injury

| | |
|---|---|
| 850.1–850.5, 850.9<br>851.0–854.1 | Intracranial injury, including concussion, confusion, laceration and haemorrhage |
| 959.01 | Head injury, unspecified |

Inclusion criteria for the matched controls will be an absence of any electronic medical history of TBI (any of the ICD9/10 codes for TBI listed in tables 1 and 2). This study is specifically designed to look at multiple outcomes emerging following TBI and their potential interactions in contributing to neurodegenerative disease risk. As such, the only exclusion in our general population comparison cohort is known TBI history. Exclusion of controls on the basis of epilepsy, for example, risks biasing our data as we would anticipate a proportion of our TBI cohort to develop post-traumatic epilepsy. Similarly, exclusion of controls with history of stroke would bias for vascular risk factors. These matched population controls will also be identified via eDRIS. Controls will be matched on year of birth, sex and socioeconomic status. Socioeconomic status is taken from last known postcode, from which area-based socioeconomic deprivation can be calculated using the Scottish Index of Multiple Deprivation (SIMD) 2020—which takes into account income, employment, health, education, housing and crime rates of that area. The SIMD is categorised into quintiles ranging from 1 (most deprived) to 5 (most affluent).[15]

Incident neurodegenerative disease diagnoses will be assessed in the cohort. These will be identified from ICD codes from a combination of primary and contributory causes of death as listed on death record certifications, and recorded hospital attendances (SMR01), and prescription information. Mental health outcomes will be analysed, identified from mental health hospital admission data (SMR04). The most common causes of death in Scottish males and females will also be assessed. Furthermore, as there are a variety of other known modifiable risk factors for dementia, these will also be assessed in accordance with dementia outcome. Known health-related modifiable risk factors for dementia may include: hypertension, hearing loss, smoking, obesity, depression, diabetes and excessive alcohol consumption.[4]

Diagnostic brain scan images and their accompanying reports archived for research purposes will also be accessed for our cohort with a history of TBI, as well as for our control cohort. By use of each individual's unique CHI number, existing diagnostic imaging and radiological reports held within eDRIS will be accessed. We will be able to match the images back to each individual's health record data, and so can make appropriate statistical adjustments for competing health outcomes. We will control for common psychological (eg, depression) and cardiometabolic conditions in fully adjusted models (eg, coronary heart disease, hypertension, diabetes, stroke) based on ICD codes. We will interrogate the routine brain scan images via standardised (semi)automated processing methods, specifically MRI and CT scans, to identify structural brain differences in individuals with dementia who have been exposed to TBI with those who have no documented medical history of TBI. One example of a structural brain change which may be seen is abnormalities of the septum pellucidum. The septum pellucidum is a thin membrane located deep in the centre of the brain, and identification of abnormalities of this structure may be one way in which we can identify neurodegeneration as a result of exposure to brain injury. Brain scans will be quantitatively assessed (eg, using the FMRIB Software Library (FSL)) and incorporate multiple brain imaging metrics of relevance to healthy ageing and dementia including hippocampal and associated subcortical phenotypes, total grey/white matter volumes and aspects of cerebrovascular health (eg, white matter hyperintensities).[16] Assessing white matter structural abnormalities (eg) may aid in establishing a potential diagnostic utility in distinguishing dementia in those with a history of TBI, and those with no history of TBI. Measures of overall brain age using existing pipelines for research analysis of MRI studies will be employed, comparing TBI to non-TBI individuals, with a main goal of finding an imaging biomarker of adverse brain health after TBI.[17] [18] Regarding availability of suitable imaging studies, the Scottish Medical Imaging Initiative includes diagnostic brain imaging from the years 2010 to 2019. In a previous study of 7676 former soccer players and their 23 028 matched general population controls the SMI identified images from over 2000 individuals in the cohort. As such, we are confident there will be adequate imaging studies available for our proposed work.

Prior to the return of the datasets from eDRIS, all individuals in the cohort will be assigned a unique identifier number, which does not enable their identity to be traced. Health record data will be anonymised by removal of name, CHI number and date of birth from all individuals in the cohort, with date of birth being replaced by month and year of birth only. Access to the excel files containing the datasets will be password protected and accessible via the National SafeHaven by named researchers only.

### Patient and public involvement

Patients will not be actively involved in this study. However, as the study progresses and data emerge, representative patient and carer organisations, such as the Scottish Acquired Brain Injury Network, the charity Headway and Brain Health Scotland, will be informed of study outcomes via our already developed public engagement pathways. As well as updating these organisations with our findings, we will ask them to promote and share these findings widely within their community.

### Statistical analyses

All statistical analyses will be undertaken on statistical software available in the National SafeHaven. For the majority of analyses, a stratified Cox proportional hazard model will be used to assess the association between TBI and risk of diseases of interest—namely neurodegenerative diseases alongside common causes of death for adults living in Scotland, as well as common mental health outcomes. Not only is TBI a risk for neurodegenerative disease, it may also be a consequence of or herald neurodegenerative disease. As such, our data will include time from TBI to measured outcomes, with the time-dependent

relationships independently analysed. A stratified model will be used to take into account the control matched nature of the data, whereby we will conduct stratified analyses based on interval. When assessing mortality data only, a competing-risks regression analysis will be undertaken to ascertain whether the estimated HR is sensitive to the competing risks of other non-neurodegenerative related deaths. There is a possibility of prodromal cognitive problems prior to any clinical dementia diagnosis, and this may result in some reverse causality, for example, increased risk of falls. We have access to data regarding time from TBI to dementia outcome, which will, however, be assessed. Missing data is a commonality in epidemiological research, and we will use three fundamental approaches to this: where appropriate, we will acknowledge this as a limitation in each output, impute based on other variables or replace with mean values. As sensitivity analyses, we will do all three and note substantial differences in results.

## Strengths and limitations

HEAD-TBI will provide novel insights into the relationship between TBI and neurodegenerative disease risk with potential to inform strategies to identify those at greatest risk of adverse outcome and possibilities for intervention to mitigate risk. TBI is recognised as a major risk factor for dementia, however, this study will further recognise the importance of greater understanding the other lifelong health outcomes following TBI—many of which may also result in increased risk of dementia. In addition, by providing unique opportunity to study trajectory of disease from initiation to outcome, HEAD-TBI will have potential wider application to research in non-TBI-associated dementias.

It is noted that although we have access to data regarding age, sex and socioeconomic status of our cohort, we have limited access to information on confounders beyond this. It is known that family history and genetics are non-modifiable risk factors for neurodegenerative diseases, which cannot be taken into account in this study. We also do not have access to any general practitioner (GP) notes or any self-report information from participants themselves. Furthermore, the NHS was not founded until 1948, and we, therefore, may have less complete medical records or no medical records at all for individuals born in the early 1900s. There are also restrictions regarding the diagnostic brain scan images, with data only available between the years 2010 and 2019. One limitation is the relative inadequacy of routine electronic health records data to categorise TBI severity. Nevertheless, we will be able to identify individuals with purely mild TBI/concussion from relevant ICD codes and also to subcategorise for TBI pathologies, such as intracranial haemorrhage, parenchymal injury and diffuse injuries, as surrogates for moderate/severe injury. Furthermore, the health record data only takes into account individuals who have been hospitalised with a TBI. This study does not consider patients managed by GPs or community based nursing staff, and therefore, may not take into consideration patients suffering from concussion/brain injury not requiring secondary care. Most likely, mild TBI, mid-moderate TBI and many psychiatric diagnoses which do not require hospitalisation will be overlooked, it is therefore considered that this analysis cannot be generalised to mild TBI and outpatient psychiatric conditions. We further recognise there may be inconsistent reporting of ICD codes however we have no reason to believe there would be any greater inconsistency or systematic bias in recording codes in those who suffered a TBI, and individuals who have not.

## Ethics and dissemination

Approvals for the study have been obtained from the University of Glasgow College of Medical, Veterinary and Life Sciences Research Ethics Committee (project number 200220038) and from NHS Scotland's Public Benefits and Privacy Panel (application 2122-0224). We would anticipate the production of multiple high impact research papers resulting from this work, and we will showcase key results to other researchers in the field, as well as to individuals living with dementia, and care workers in the field of dementia. The interrogation of diagnostic images for abnormalities of brain structure may help us distinguish neurodegenerative diseases resultant from a history of TBI and may have diagnostic utility. This work has the potential to inform clinical practice, and be of benefit to dementia researchers, clinicians and individuals suffering with dementia.

**Contributors** ERR produced the first draft of the manuscript; DML and WS are involved in supervisory oversight for the HEAD-TBI study, and also codesigned the study protocol and edited the manuscript.

**Funding** The Chief Scientist Office (CSO), Award Number: EPD/22/07.

**Competing interests** None declared.

**Patient and public involvement** Patients and/or the public were not involved in the design, or conduct, or reporting, or dissemination plans of this research.

**Patient consent for publication** Not applicable.

**Provenance and peer review** Not commissioned; externally peer reviewed.

**ORCID iDs**
Emma Rosalyn Russell http://orcid.org/0000-0002-5320-5910
William Stewart http://orcid.org/0000-0003-2199-2582

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
