## [Reviewer comments · BMJ Open]

ARTICLE DETAILS

TITLE (PROVISIONAL)	HEalth And Dementia outcomes following Traumatic Brain Injury (HEAD-TBI): protocol for a retrospective cohort study
AUTHORS	Russell, Emma; Lyall, Donald; Stewart, William

VERSION 1 – REVIEW

REVIEWER	Schaffert, Jeff The University of Texas Southwestern Medical Center
REVIEW RETURNED	13-Apr-2023

GENERAL COMMENTS	Overall strengths: The rationale for the protocol is well laid out and easy to digest. Additionally, the overall reasoning for the direction of identifying the target population appears to be straightforward and hopefully relatively easy to accomplish. Areas of Concern: The protocol is essentially an outline of the methods and is lacking in significant detail. Additional areas that need to be clarified include: - time frame from TBI (is there a cutoff?). This is very important because we know neurodegenerative processes begin many years before the onset of symptoms. As such, the having a clear cut-off from TBI to ICD diagnoses of dementia is needed.- The protocol's rationale for including a matched control group makes sense, but the protocol would benefit from additional explanation of how these controls are selected. For example, exclusion criteria (e.g. history of stroke, epilepsy, etc.) are completely absent.- What are the specific outcomes you are going to evaluate? These are mentioned but need more detail. Again, how will the timing in relation to TBI be defined for these additional outcomes.- Much more detail is needed on radiology reports. What are the variables of interest besides septum pellucidum? Are you going to evaluate hippocampal volumes (if so, any adjustment to age)? How will this be defined? Manually or quantitatively? I was also wondering about the prevalence of diagnostic brain scan images in both the target population and the matched controls - will these be widely available in matched controls?- Any covariates for the Cox analyses?
---

	- What are the investigators going to do with inconsistency in ICD codes? - How will dementia be defined? ICD codes or causes of death. This is not clear. - How will TBI severity be stratified? - In general, there needs to be more explanation on why this is novel. The literature has had many health review record studies examining ICD codes. What is novel about this study specifically? A quick review of the literature has the following with similar designs (none of which are cited in the background): Lee, Y. K., Hou, S. W., Lee, C. C., Hsu, C. Y., Huang, Y. S., & Su, Y. C. (2013). Increased risk of dementia in patients with mild traumatic brain injury: a nationwide cohort study. PLoS One, 8(5), e62422. doi:10.1371/journal.pone.0062422 Gardner, R. C., Burke, J. F., Nettiksimmons, J., Kaup, A., Barnes, D. E., & Yaffe, K. (2014). Dementia risk after traumatic brain injury vs nonbrain trauma: the role of age and severity. JAMA Neurol, 71(12), 1490-1497. doi:10.1001/jamaneurol.2014.2668 Barnes, D. E., Byers, A. L., Gardner, R. C., Seal, K. H., Boscardin, W. J., & Yaffe, K. (2018). Association of Mild Traumatic Brain Injury With and Without Loss of Consciousness With Dementia in US Military Veterans. JAMA Neurol, 75(9), 1055-1061. doi:10.1001/jamaneurol.2018.0815 Nordstrom, A., & Nordstrom, P. (2018). Traumatic brain injury and the risk of dementia diagnosis: A nationwide cohort study. PLoS Med, 15(1), e1002496. doi:10.1371/journal.pmed.1002496 Osler, M., Rozing, M. P., Eliassen, M. H., Christensen, K., & Mortensen, E. L. (2020). Traumatic brain injury and risk of dementia at different levels of cognitive ability and education. Eur J Neurol, 27(2), 399-405. doi:10.1111/ene.14095 Schneider, A. L. C., Selvin, E., Latour, L., Turtzo, L. C., Coresh, J., Mosley, T., . . . Gottesman, R. F. (2021). Head injury and 25-year risk of dementia. Alzheimers Dement, 17(9), 1432-1441. doi:10.1002/alz.12315
--	--

REVIEWER	Kenney, Kimbra Walter Reed National Military Medical Center, Department of Neurology
REVIEW RETURNED	18-Apr-2023

GENERAL COMMENTS	This is a relatively brief overview of the funded study that is currently underway, but needs more details filled in to determine how the authors are going to determine risk factors for dementia and neurodegenerative conditions among individuals who've had a lifetime TBI compared to those who haven't. I suggest the authors revise their manuscript to include the following: 1) Add estimate of how many TBI patients they project will be in the database and the expected TBI severity breakdown. As the authors are well aware, TBI dxes and TBI diagnostic criteria have changed dramatically over the past 100 years and before 2000, it is unlikely
---

	that mTBI will be diagnosed, despite currently accounting for 85-90% of all TBIs. Currently, it is estimated that about 30% of the population or greater will suffer a lifetime TBI. The epidemiologic association between mTBI and dementia and neurodegenerative conditions is controversial with mixed results from large epidemiologic conditions. It is not clear how the team will account for potentially undiagnosed lifetime mTBI in the 3:1 matched controls for TBI patients before 2000 as their medical records very well may not have that diagnosis and they may be erroneously used as a control. At the very least, this needs to be added to the limitations, but more importantly, considered in study design and selection of control participants. How will they account for TBI that does/did not require hospitalization or didn't have any imaging? 2) The methods section needs to clarify how dementia will be diagnosed and distinguished from static encephalopathy or secondary encephalopathy from chronic medical illness (e.g. renal disease). They also need to clarify how a variety of neurodegenerative disorders will be diagnosed- by neuropathological exam or clinician dx in the EHR only. As, the authors noted, EHR diagnoses are fraught with errors and omissions and are not reliable. 3) How will the authors account for conditions that may be indirectly linked to death, but associated with dementia/neurodegenerative disorders and TBI, e.g. OSA, diabetes, chronic tobacco use and hypertension. Chronic illnesses are not always first noted when they first start. 4) Further, how will they determine the timing of disease onset/diagnosis and determine association/causality.... E.g. which occurred first, the DM or the dementia? Or the alcohol use and the TBI? It could be that alcohol use is a risk factor for TBI rather than than a history of TBI is a risk factor for alcohol abuse. 5) Should PTSD and suicidality (there is an ICD-10 code for that) be included in the list of psychiatric illnesses? The CDC includes suicide by gun to their national TBI incidence database. How will the authors account for these behavioral conditions in their risk analysis and, in particular, how will they determine which occurred first. 6) For the imaging assessments, will they include controls with imaging and which imaging from TBI patients will they use to include in their analysis (acute, subacute, chronic)? Will they compare imaging from dementia without lifetime TBI to dementia patients with lifetime TBI and will they age match the images? How will they control for other conditions that are associated with WM changes, like hypertension, OSA, DM? How will they control for age and time since injury? How will they control for much earlier generation CT vs MRI? 7) Since they will use death certificate date, will they have access to autopsy information include neuropathology and how will they use that or is it too rare to be helpful in confirming clinical neurodegenerative diagnoses, such as Alzheimer's or Parkinson's to be used in the analysis 8) Finally, the statistical section needs to be expanded and described more in detail, including how missing data will be dealt with, as well as how the limitations noted in the paper, as well as some of those listed above will be handled. 9) Finally, please add a consort diagram to the manuscript as well as a table of inclusion/exclusion criteria
--	--

VERSION 1 – AUTHOR RESPONSE

Reviewer 1:

The rationale for the protocol is well laid out and easy to digest. Additionally, the overall reasoning for the direction of identifying the target population appears to be straightforward and hopefully relatively easy to accomplish.

We thank the reviewer for this comment.

1. The protocol is essentially an outline of the methods and is lacking in significant detail. Additional areas that need to be clarified include: - time frame from TBI (is there a cutoff?). This is very important because we know neurodegenerative processes begin many years before the onset of symptoms. As such, the having a clear cut-off from TBI to ICD diagnoses of dementia is needed.

We are in complete agreement with the Reviewer over the need for data on time from TBI to outcome in our analysis.

The following text has been added to the statistical analysis section of the manuscript: *“Not only is TBI a risk for neurodegenerative disease, it may also be a consequence of or herald neurodegenerative disease. As such, our data will include time from TBI to measured outcomes, with the time-dependent relationships independently analysed.”*

2. The protocol's rationale for including a matched control group makes sense, but the protocol would benefit from additional explanation of how these controls are selected. For example, exclusion criteria (e.g. history of stroke, epilepsy, etc.) are completely absent.

The following text has been added to the methods section of the manuscript:

“This study is specifically designed to look at multiple outcomes emerging following TBI and their potential interactions in contributing to neurodegenerative disease risk. As such, the only exclusion in our general population comparison cohort is known TBI history. Exclusion of controls on the basis of epilepsy, by example, risks biasing our data as we would anticipate a proportion of our TBI cohort to develop post-traumatic epilepsy. Similarly, exclusion of controls with history of stroke would bias for vascular risk factors.”

3. What are the specific outcomes you are going to evaluate? These are mentioned but need more detail. Again, how will the timing in relation to TBI be defined for these additional outcomes.

The following text has been added to the methods section to give additional detail:

“Incident neurodegenerative disease diagnoses will be assessed in the cohort. These will be identified from ICD codes from a combination of primary and contributory causes of death as listed on death record certifications, and recorded hospital attendances (Scottish Morbidity Record (SMR) 01), and prescription information. Mental health outcomes will be analysed, identified from mental health hospital admission data (SMR04). The most common causes of death in Scottish males and females will also be assessed. Furthermore, as there are a variety of other known modifiable risk factors for dementia, these will also be assessed in accordance with dementia outcome.”

4. Much more detail is needed on radiology reports. What are the variables of interest besides septum pellucidum? Are you going to evaluate hippocampal volumes (if so, any adjustment to age)? How will this be defined? Manually or quantitatively? I was also wondering about the prevalence of diagnostic brain scan images in both the target population and the matched controls - will these be widely available in matched controls?

We apologise for the lack of clarity over our proposed imaging studies and have amended the manuscript to include information on our proposed analysis as:

“Scans will be quantitatively assessed (e.g. using FSL) and incorporate multiple brain imaging metrics of relevance to healthy ageing and dementia including hippocampal and associated subcortical phenotypes, total grey/white matter volumes and aspects of cerebrovascular health (e.g. white matter hyperintensities) ...

Regarding availability of suitable imaging studies, the Scottish Medical Imaging Initiative includes diagnostic brain imaging from the years 2010 to 2017. In a previous study of 7,676 former soccer players and their 23,028 matched general population controls the SMI identified images from over 2,000 individuals in the cohort. As such, we are confident there will be adequate imaging studies available for our proposed work.”

5. Any covariates for the Cox analyses?

Year of birth, age, and sex have already been taken into consideration through the matching of 3 controls. The following has been added to the statistical analyses paragraph:

“When assessing mortality data only, a competing-risks regression analysis will be undertaken to ascertain whether the estimated hazard ratio is sensitive to the competing risks of other non-neurodegenerative related deaths.”

6. What are the investigators going to do with inconsistency in ICD codes?

We recognise that this is an important issue, the following text has been added to the strengths and limitations section to clarify:

“We further recognise there may be inconsistent reporting of ICD codes however we have no reason to believe there would be any greater inconsistency or systematic bias in recording codes in those who suffered a TBI, and individuals who have not.”

7. How will dementia be defined? ICD codes or causes of death. This is not clear.

The following text has been added to the methods to give additional detail:

“Incident neurodegenerative disease diagnoses will be assessed in the cohort. These will be identified from ICD codes from a combination of primary and contributory causes of death as listed on death record certifications, and recorded hospital attendances (Scottish Morbidity Record (SMR) 01), and prescription information.”

8. How will TBI severity be stratified?

The reviewer raises an important observation in respect of this work. We have added text to our limitation section as:

“One limitation is the relative inadequacy of routine electronic health records data to categorise TBI severity. Nevertheless, we will be able to identify individuals with purely mild TBI/ concussion from relevant ICD codes and also to subcategorise for TBI pathologies, such as intracranial hemorrhage, parenchymal injury and diffuse injuries, as surrogates for moderate/severe injury.”

9. In general, there needs to be more explanation on why this is novel. The literature has had many health review record studies examining ICD codes. What is novel about this study specifically? A quick review of the literature has the following with similar designs (none of which are cited in the background):

The following text has been added to the strengths and limitations section to give additional detail:

“HEAD-TBI will provide novel insights into the relationship between TBI and neurodegenerative disease risk with potential to inform strategies to identify those at greatest risk of adverse outcome and possibilities for intervention to mitigate risk. Traumatic brain injury (TBI) is recognised as a major risk factor for dementia, however this study will further recognise the importance of greater understanding the other lifelong health outcomes following TBI – many of which may also result in increased risk of dementia. In addition, by providing unique opportunity to study trajectory of disease from initiation to outcome, HEAD-TBI will have potential wider application to research in non-TBI associated dementias.”

Moreover, the following references have been inserted in the main body of text: Lee, Y. K., Hou, S. W., Lee, C. C., et al. (2013). Increased risk of dementia in patients with mild traumatic brain injury: a nationwide cohort study. *PLoS One*, 8(5), e62422. doi:10.1371/journal.pone.0062422

Gardner, R. C., Burke, J. F., Nettiksimmons, J., et al. (2014). Dementia risk after traumatic brain injury vs nonbrain trauma: the role of age and severity. *JAMA Neurol*, 71(12), 1490-1497. doi:10.1001/jamaneurol.2014.2668

Schneider, A. L. C., Selvin, E., Latour, L., et al. (2021). Head injury and 25-year risk of dementia. *Alzheimers Dement*, 17(9), 1432-1441. doi:10.1002/alz.12315

Nordstrom, A., & Nordstrom, P. (2018). Traumatic brain injury and the risk of dementia diagnosis: A nationwide cohort study. *PLoS Med*, 15(1), e1002496. doi:10.1371/journal.pmed.1002496

Elser H, Gottesman RF, Walter AE, et al. Head Injury and Long-term Mortality Risk in Community-Dwelling Adults. *JAMA Neurol*. 2023;80(3):260.

Reviewer 2:

This is a relatively brief overview of the funded study that is currently underway, but needs more details filled in to determine how the authors are going to determine risk factors for dementia and neurodegenerative conditions among individuals who've had a lifetime TBI compared to those who haven't. I suggest the authors revise their manuscript to include the following:

1. Add estimate of how many TBI patients they project will be in the database and the expected TBI severity breakdown. As the authors are well aware, TBI dxes and TBI diagnostic criteria have changed dramatically over the past 100 years and before 2000, it is unlikely that mTBI will be diagnosed, despite currently accounting for 85-90% of all TBIs. Currently, it is estimated that about 30% of the population or greater will suffer a lifetime TBI. The epidemiologic association between mTBI and dementia and neurodegenerative conditions is controversial with mixed results from large epidemiologic conditions. It is not clear how the team will account for potentially undiagnosed lifetime mTBI in the 3:1 matched controls for TBI patients before 2000 as their medical records very well may not have that diagnosis and they may be erroneously used as a control. At the very least, this needs to be added to the limitations, but more importantly, considered in study design and selection of control participants. How will they account for TBI that does/did not require hospitalization or didn't have any imaging?

We agree that this is an important limitation, whilst our data will capture the more severe end of health outcomes, it may miss some milder symptoms. We have now included in the limitations section of the manuscript:

"...the health record data only takes into account individuals who have been hospitalised with a traumatic brain injury. This study does not consider patients managed by general practitioners or community based nursing staff, and therefore may not take into consideration patients suffering from concussion / brain injury not requiring secondary care."

2. The methods section needs to clarify how dementia will be diagnosed and distinguished from static encephalopathy or secondary encephalopathy from chronic medical illness (e.g. renal disease). They also need to clarify how a variety of neurodegenerative disorders will be diagnosed- by neuropathological exam or clinician dx in the EHR only. As, the authors noted, EHR diagnoses are fraught with errors and omissions and are not reliable.

Please see response to Reviewer 1 question 7 regarding ascertainment of incident dementia diagnoses. The Reviewer is correct to highlight recognised inaccuracies in electronic health records data. However, our analysis considers relative risk of outcome in TBI exposed versus their matched controls, not absolute numbers affected. Further, we have no reason to believe any systematic bias to over or under reporting of outcomes in study group.

3. How will the authors account for conditions that may be indirectly linked to death, but associated with dementia/neurodegenerative disorders and TBI, e.g. OSA, diabetes, chronic tobacco use and hypertension. Chronic illnesses are not always first noted when they first start.

This an important point. When analysing neurodegenerative diseases as a cause of death, we will adjust for competing risks, such as hypertension etc. Whilst chronic illnesses may not always receive a formal diagnosis initially, we will have access to datasets such as prescription information and inpatient hospital attendances, which will likely capture diseases at the earlier stage.

The following text has been added to the statistical analysis section to clarify:

“When assessing mortality data only, a competing-risks regression analysis will be undertaken to ascertain whether the estimated hazard ratio is sensitive to the competing risks of other non-neurodegenerative related deaths.”

4. Further, how will they determine the timing of disease onset/diagnosis and determine association/causality.... E.g. which occurred first, the DM or the dementia? Or the alcohol use and the TBI? It could be that alcohol use is a risk factor for TBI rather than than a history of TBI is a risk factor for alcohol abuse.

Time at event is included in the dataset. We will have access to the date of every hospital admission/prescription etc. for every disease of interest. The following text has been added to the manuscript to clarify:

“Not only is TBI a risk for neurodegenerative disease, it may also be a consequence of or herald neurodegenerative disease. As such, our data will include time from TBI to measured outcomes, with the time-dependent relationships independently analysed.”

5. Should PTSD and suicidality (there is an ICD-10 code for that) be included in the list of psychiatric illnesses? The CDC includes suicide by gun to their national TBI incidence database. How will the authors account for these behavioral conditions in their risk analysis and, in particular, how will they determine which occurred first.

Suicidality will be captured under our proposed codes for mental health outcomes. Again, dates for each record of outcome of interest will be available, allowing the time dependent interactions in outcomes to be addressed.

6. For the imaging assessments, will they include controls with imaging and which imaging from TBI patients will they use to include in their analysis (acute, subacute, chronic)? Will they compare imaging from dementia without lifetime TBI to dementia patients with lifetime TBI and will they age match the images? How will they control for other conditions that are associated with WM changes, like hypertension, OSA, DM? How will they control for age and time since injury? How will they control for much earlier generation CT vs MRI?

Additional details have been added to the methods section to clarify:

“Diagnostic brain scan images and their accompanying reports archived for research purposes will also be accessed for our cohort with a history of TBI, as well as for our control cohort ...

Brain scans will be quantitatively assessed (e.g. using FSL) and incorporate multiple brain imaging metrics of relevance to healthy ageing and dementia including hippocampal and associated subcortical phenotypes, total grey/white matter volumes and aspects of cerebrovascular health (e.g. white matter hyperintensities) ...

Regarding availability of suitable imaging studies, the Scottish Medical Imaging Initiative includes diagnostic brain imaging from the years 2010 to 2017. In a previous study of 7,676 former soccer players and their 23,028 matched general population controls the SMI identified images from over

2,000 individuals in the cohort. As such, we are confident there will be adequate imaging studies available for our proposed work.”

7. Since they will use death certificate date, will they have access to autopsy information include neuropathology and how will they use that or is it too rare to be helpful in confirming clinical neurodegenerative diagnoses, such as Alzheimer’s or Parkinson’s to be used in the analysis.

The Reviewer raises an excellent observation. However, regrettably, the electronic health records are not linked to autopsy reporting datasets precluding this ideal analysis.

8. Finally, the statistical section needs to be expanded and described more in detail, including how missing data will be dealt with, as well as how the limitations noted in the paper, as well as some of those listed above will be handled.

Thank you for this suggestion. The following text has been added to the statistical analyses paragraph:

“Missing data is a commonality in epidemiological research with three fundamental approaches: acknowledge as a limitation, impute based on other variables, or replace with mean values. As sensitivity analyses we will do all three and note substantial differences in results.”

9. Finally, please add a consort diagram to the manuscript as well as a table of inclusion/exclusion criteria.

As this study is not a randomised controlled trial and we do not yet have access to the datasets which will be analysed, a consort diagram is proving difficult to produce. We have instead inserted a RECORD checklist at the end of the manuscript, and have included further details regarding inclusion/exclusion criteria within the main body of the text.

VERSION 2 – REVIEW

REVIEWER	Kenney, Kimbra Walter Reed National Military Medical Center, Department of Neurology
REVIEW RETURNED	07-Jun-2023

GENERAL COMMENTS	Thank you to the authors for their thorough and thoughtful consideration of this reviewer's suggested edits. The majority have been incorporated into the manuscript and it is nearly ready for publication. A few remaining issues need to be included, as follows: 1) Please add a list of health outcomes/outcomes that will be included in the statistical models that the authors state are also associated with TBI (e.g. which cardiovascular and psychiatric diseases- e.g. diabetes, hypertension, hyperlipidemia, etc.) 2) Please add a discussion of how the "chicken/egg" issue will be addressed in the analysis- that is dementia/neurodegenerative disorder is a risk factor for TBI (through falls) and perhaps an even greater risk factor than early/mid life TBI is a risk factor for later life dementia/neurodegenerative disorder. 3) The control group will be age, sex and social/economically matched. How will the images be matched? Will you only compare images to those who have dementia in each group and look at the differences in dementia scans in those with compared to without TBI to see how they differ? WMHs CAN be caused by TBI, but they are more likely caused by CVD, especially in the elderly, and how will non-dementia outcomes be assessed in the images. Further, there is a burgeoning literature on imaging biomarkers of psychiatric
--

	disorders, particularly focused on structural changes in the limbic system. Please add further discussion as to how the images will be assessed, including controlling for other diagnoses that may be equally associated with imaging changes of interest in TBI-associated dementia. Finally, with regard to imaging, will the study team only use advanced MRI or how will it compare volume collected by MRI versus CT across subjects? 4) Since only ICD coding will be used for both life time TBI and psychiatric diagnoses, please add a statement to the limitations that, most likely life time mTBI and mod-moderate, (not requiring hospitalization) psychiatric diagnoses will not be adequately collected and that this analysis cannot therefore be generalized to mTBI and outpatient psychiatric conditions. Consider a change to the title emphasizing that only TBI and psychiatric diagnoses requiring hospitalization will be included in the study (e.g. "HEalth and Dementia outcomes following TBI (HEAD-TBI) requiring hospitalization: protocol for a...."). 5) Will the analysis look at a dose effect of TBI? While (hopefully) the majority of individuals only suffer a single life time severe TBI, many individuals have multiple, even numerous, life time mTBI/concussions. Will the analysis assess a life time TBI dose effect? If not, please add a discussion to the limitation section regarding the potential chronic effects of repetitive mild TBI and the development of dementia and that this analysis will not likely capture the controversial association.
--	---

VERSION 2 – AUTHOR RESPONSE

Thank you to the authors for their thorough and thoughtful consideration of this reviewer's suggested edits. The majority have been incorporated into the manuscript and it is nearly ready for publication. A few remaining issues need to be included, as follows:

1. Please add a list of health outcomes/outcomes that will be included in the statistical models that the authors state are also associated with TBI (e.g. which cardiovascular and psychiatric diseases- e.g. diabetes, hypertension, hyperlipidemia, etc.)

We have added additional detail to the methods section to clarify this:

*“The most common causes of death in Scottish males and females will also be assessed. Furthermore, as there are a variety of other known modifiable risk factors for dementia, these will also be assessed in accordance with dementia outcome. **Known health related modifiable risk factors for dementia may include: hypertension, hearing loss, smoking, obesity, depression, diabetes, and excessive alcohol consumption** ⁴.”*

2. Please add a discussion of how the "chicken/egg" issue will be addressed in the analysis-that is dementia/neurodegenerative disorder is a risk factor for TBI (through falls) and perhaps an even greater risk factor than early/mid life TBI is a risk factor for later life dementia/neurodegenerative disorder.

We have amended the manuscript to note:

“There is a possibility of prodromal cognitive problems prior to any clinical dementia diagnosis, and this may result in some reverse causality, e.g. increased risk of falls. We have access to data regarding time from TBI to dementia outcome, which will, however, be assessed.”

3. The control group will be age, sex and social/economically matched. How will the images be matched? Will you only compare images to those who have dementia in each group and look at the differences in dementia scans in those with compared to without TBI to see how they differ? WMHs CAN be caused by TBI, but they are more likely caused by CVD, especially in the elderly, and how will non-dementia outcomes be assessed in the images. Further, there is a burgeoning literature on imaging biomarkers of psychiatric disorders, particularly focused on structural changes in the limbic system. Please add further discussion as to how the images will be assessed, including controlling for other diagnoses that may be equally associated with imaging changes of interest in TBI-associated dementia. Finally, with regard to imaging, will the study team only use advanced MRI or how will it compare volume collected by MRI versus CT across subjects?

We have added additional detail to the imaging paragraph to clarify:

“We will be able to match the images back to each individual’s health record data, and so can make appropriate statistical adjustments for competing health outcomes. We will control for common psychological (e.g. depression) and cardiometabolic conditions in fully-adjusted models (e.g. Coronary heart disease; hypertension; diabetes; stroke) based on ICD codes. ...

Measures of overall brain age using existing pipelines for research analysis of MRI studies will be employed, comparing TBI to non-TBI individuals, with a main goal of finding an imaging biomarker of adverse brain health after TBI.”

4. Since only ICD coding will be used for both life time TBI and psychiatric diagnoses, please add a statement to the limitations that, most likely life time mTBI and mod-moderate, (not requiring hospitalization) psychiatric diagnoses will not be adequately collected and that this analysis cannot therefore be generalized to mTBI and outpatient psychiatric conditions. Consider a change to the title emphasizing that only TBI and psychiatric diagnoses requiring hospitalization will be included in the study (e.g. "HEalth and Dementia outcomes following TBI (HEAD-TBI) requiring hospitalization: protocol for a....".

We have added additional detail to the limitations section to clarify this:

“This study does not consider patients managed by general practitioners or community based nursing staff, and therefore may not take into consideration patients suffering from concussion / brain injury not requiring secondary care. Most likely, mild TBI, mid-moderate TBI, and many psychiatric diagnoses which do not require hospitalization will be overlooked, it is therefore considered that this analysis cannot be generalized to mild TBI and outpatient psychiatric conditions.”

HEAD-TBI is the approved study title, however information on data source can and will be added to any results papers and reflected in appropriate titles.

5. Will the analysis look at a dose effect of TBI? While (hopefully) the majority of individuals only suffer a single life time severe TBI, many individuals have multiple, even numerous, life time mTBI/concussions. Will the analysis assess a life time TBI dose effect? If not, please add

a discussion to the limitation section regarding the potential chronic effects of repetitive mild TBI and the development of dementia and that this analysis will not likely capture the controversial association.

We thank the reviewer for this insightful comment. Where applicable participants with 1/2/3 (etc.) records of TBI prior to dementia will be reported, and dose-response analyses conducted should numbers permit. We note the comment on perceived limitation regarding repetitive mild TBI. Our parallel studies on former contact sport athletes are more appropriate to addressing these issues, with this current work solely focussed on health record recording of diagnosed TBI.

VERSION 3 – REVIEW

REVIEWER	Kenney, Kimbra Walter Reed National Military Medical Center, Department of Neurology
REVIEW RETURNED	26-Jun-2023
GENERAL COMMENTS	Thank you to the authors for their careful review of my comments/suggestions and revision of the manuscript. I have no further comments/concerns.